

# A new web-based system to improve the monitoring of snow avalanche hazard in France

Ekaterina Bourova[1], Eric Maldonado[1], Jean-Baptiste Leroy[2], Rachid Alouani[2], Nicolas Eckert[2], Mylene Bonnefoy-Demongeot[2], and Michael Deschatres[2]

[1]SGGR, IRSTEA/Université Grenoble Alpes, BP 76, 38402 Saint Martin d'Hères, France
[2]UR ETGR, IRSTEA/Université Grenoble Alpes, BP 76, 38402 Saint Martin d'Hères, France

*Correspondence to:* Ekaterina Bourova (bourova@gmail.com)

**Abstract.** Snow avalanche data in the French Alps and Pyrenees have been recorded for more than 100 years in several databases. The increasing amount of observed data required a more integrative and automated service. Here we report the comprehensive web-based Snow Avalanche Information System newly developed to this end for three important datasets: an avalanche chronicle (Enquête
Permanente sur les Avalanches, EPA), an avalanche map (Carte de Localisation des Phénomènes d'Avalanche, CLPA) and a compilation of hazard and vulnerability data recorded on selected paths endangering human settlements (Sites Habités Sensibles aux Avalanches, SSA). These datasets are now integrated into a common database, enabling full interoperability between all different types of snow avalanche records: digitized geographic data, avalanche descriptive parameters, eyewitness re-
ports, photographs, hazard and risk levels, etc. The new information system is implemented through modular components using Java-based web technologies with Spring and Hibernate frameworks. It automates the manual data entry and improves the process of information collection and sharing, enhancing user experience, data quality, and offering new outlooks to explore and exploit the huge amount of snow avalanche data available for fundamental research and more applied risk assess-
ment.

## 1 Introduction

Highly populated mountain regions in the world are exposed to an important risk of snow avalanches. To assess avalanche hazard in these areas, regular snow avalanche observations are essential. The information about avalanches is often difficult to obtain due to hard field work conditions, especially
in remote zones. Even though the observed data are generally imperfect, they remain very useful for avalanche risk prevention (Tacnet et al., 2014). Moreover, avalanche observations provide important data for scientific research devoted to the development of better physical models of avalanche release and propagation, risk analysis, as well as the design of protection structures (e.g. Ancey et al., 2004; Naaim et al., 2004; Eckert et al., 2009). Several authors also show the importance of sufficiently
long data series to study the variation of natural avalanche activity in the context of climate change



(e.g. Laternser and Schneebeli, 2002; Eckert et al., 2010a, c; Corona et al., 2012; Castebrunet et al., 2014).

Most of the countries exposed to avalanche risk perform a regular registration of avalanche activity (Bonnefoy-Demongeot et al., 2014). In Switzerland, avalanche data are systematically collected since 1950 by the Swiss Federal Institute for Snow and Avalanche Research (SLF). Two types of data are stored: the Avalanche Observations (AO) which provide data of general avalanche activity (number, size and impact), including the specification of the triggering mechanism; and the Destructive Avalanches Database (DADB) covering more than 10 000 individual avalanches having caused property damage or affected people (Laternser and Schneebeli, 2002).

Avalanche data in United States were initially compiled into avalanche atlases at local scale (Frutiger, 1964; Miller et al., 1976; Armstrong and Armstrong, 1977, 2006). A generalized long-term Westwide Avalanche Network (WWAN) database of weather, snowpack and avalanche information is available since 1967 (Williams, 1994; Tremper, 1996). The WWAN database contains records for both natural and controlled avalanches, providing daily data on the number and size of avalanches (Birkeland and Mock, 2001). In Canada, the Information Exchange (InfoEx) database of the Canadian Avalanche Association (CAA) provides daily avalanche occurrences since 1991 submitted by avalanche industry professionals (Haegeli and McClung, 2007; Haegeli et al., 2014). In Europe, the observational, topographical and meteorological data for extreme or deflected avalanche events are available in the EU CADZIE database (Domaas et al., 2002). A large campaign of avalanche observations was conducted in the former Soviet Union from 1974 to 1991. These observations were compiled into the Avalanche Cadastre of the USSR divided into 20 volumes and covering wide areas of important avalanche activity (Gidrometeoizdat, 1984-1991).

Several databases provide geolocalized information about avalanche paths. In the Catalan Pyrenees, avalanche data are stored into the Avalanche Database of Catalonia (ADBC) with more than 17 000 mapped avalanche paths (Oller et al., 2006). In the Italian Alps, the avalanche cadastre includes 3 000 avalanche paths with accurately plotted borders (Debernardi and Segor, 2013). Spatial characteristics of approximately 2 300 avalanches were recently obtained in Japan from aerial photographs (Akiyama and Ikeda, 2013).

Current advances in computer technologies allow for using relational databases, Geographic Information System (GIS) tools and web applications to store snow avalanche observations and share them via the Internet (Tremper, 1996; Bründl et al., 2004; Scott, 2007; Duclos et al., 2008; Scott, 2009; Ekker et al., 2013). Web-based applications represent also a suitable software solution to assist the observers in their daily work of data collection and validation. They are used to support simultaneous access for a wide range of users with different roles. Such new web-based tools with computer assisted workflow have been recently implemented to support the forecasters in producing avalanche bulletins in Switzerland (Ruesch et al., 2013) and in Norway (Jaedicke et al., 2014). In



Austria and Italy, web-based platforms have been developed to support the management of local avalanche hazard (Segor et al., 2014).

In France, the first regular avalanche observations started as early as the beginning of the 20th century with the creation of the EPA observation system that describes avalanche events on a selection of paths (Deschatres et al., 2010b). On this selection, the system provides information about year-to-year avalanche activity but it is not designed to map all avalanche prone areas. In 1971, the CLPA observation system was created to fill this gap. The CLPA carries out a regular survey of the regions affected by avalanches in order to produce a map of all avalanche maximum limits (Borrel, 1992; Bonnefoy et al., 2010, 2012; Robinet and Bonnefoy-Demongeot, 2013), providing the spatial dimension to snow avalanche observations. Finally, the SSA dataset has been created to inventory particular avalanche paths endangering human settlements (Rapin et al., 2006). These three data sets are introduced in more details in Sect. 2.

For all three observation systems, the data acquisition was initially conducted via a manual process. The technical staff in charge of observations sent paper reports to a coordinator who centralized the information. Later, the data integration was partly automatized by desktop Access applications developed in Visual Basic for Applications (VBA). Also, the EPA, CLPA and SSA systems evolved independently and resulted in three completely separated datasets. However, the management of the increasing amount of available data as well as new research and operational needs that progressively emerged made the development of a new integrated information system unavoidable. It should guarantee the interoperability among the different datasets as well as a modulable multi-user and multi-level accesses. In a first attempt to this purpose, a web application has been designed for the EPA and SSA tools with a multi-user access and a common database (Vidaud-Barral et al., 2010). Yet, the CLPA was not integrated in this work because the needs were not fully defined at that time.

In this paper, we present a generalized web-based snow avalanche information system that includes all CLPA, EPA and SSA modules. We also illustrate the main technical advantages of this new system: 1) it provides a computer assisted homogeneous workflow for observers, accessible from everywhere; 2) it creates a real time multi-user collaborational platform; 3) it insures the data consistency, interoperability and security; 4) the system is built on a stable framework that can be easily maintained and can be further extended to include additional components and services. In what follows, in Sect. 2, we describe the avalanche data available in our database and the corresponding acquisition processes. In Sect. 3, we summarize the main requirements of the system and the user's needs. On this basis , we present in Sect 4. the new modular architecture that enables a full interoperability among its different components: EPA, CLPA and SSA. Finally, in Sect. 5, as an illustrative example, we detail the implementation of the web platform through key operations for the CLPA workflow.



## 2 Data and acquisition processes

The observations of avalanche phenomena in France are coordinated by the French Ministry of the Environment. The observations are systematically recorded in mountain regions of the French Alps and Pyrenees. A few events also occur in lower altitude mountain ranges, but they are currently not systematically collected by state services. The data are collected by professional staffs of the French National Forest Office and Irstea composed of approximately 300 forest rangers and avalanche practitioners. The data acquisition process follows a well-defined protocol to obtain, as much as possible, complete, regular and homogeneous datasets. All the collected data are peer-reviewed before being published.

### 2.1 Avalanche chronicle: EPA

The Avalanche Permanent Survey (EPA, Enquête Permanente sur les Avalanches) initially started in Savoie (Mougin, 1922) but has been rapidly extended to the whole French Alps and Pyrenees. Nowadays, the EPA database contains nearly 100 000 avalanche events on around 3 900 paths. A key strength of the EPA dataset consists in the fact that it provides a long temporal series of avalanche observations on well specified paths. Some of EPA paths are observed since more than 100 years.

The data are continuously updated by observers through annual campaigns that start in September and end in August. An EPA path is placed under permanent or intermittent observation. The EPA paths with easy access are permanently visited by the observer after each snowfall. For hardly accessible areas, the observer visits a path at least once in a winter. The observation of such paths is considered as intermittent.

To decide whether an occurred avalanche event is representative enough for a path, an *observation threshold* and an *alert threshold* are defined. The choice of these thresholds is based on expert judgement considering geomorphological characteristics of each site. Important avalanche events that reach the observation threshold are regularly registered into the database. Smaller avalanches stopped above the observation threshold are generally not registered. Yet, sometimes such small avalanches can also be recorded if the observer considers them significant for some reasons (important snow volume, damages, victims). Major avalanches that reach the alert threshold are recorded in the database with a special alert status. For these events, the observer has to alert the National Forest Office and Irstea within 48 hours for possible additional investigations.

The observers record the following information about an avalanche: date and time of the event or, if such detailed information is not available, at least a time frame within which the avalanche has occurred, release and runout altitudes, snow deposit volume, type of trigger, basic snow characteristics, weather conditions before the event, victims and damages. The obtained data are peer-reviewed by another observer in charge of control. At the end of a campaign, the following validated output materials are published on the web in open access: geographic maps of the EPA paths, observation



and alert thresholds, EPA path photographs and the lists of avalanche events observed on each EPA path.

An example of EPA output materials obtained for the path number 7 in Molines-en-Queyras town-
ship is presented in Figures 1 and 2. The EPA path limits, the positions of observation and alert thresholds, the talweg and the observation point (*i.e.* the exact place that the observer has to visit regularly) are plotted in Figure 1. Figure 2 shows the list of recent avalanche events recorded on this path.

### 2.2    Avalanche map: CLPA

The Localization Map of Avalanche Phenomena (CLPA) covers 345 townships and includes more than 25 000 avalanche paths in the French Alps and Pyrenees. An example of the CLPA map is shown in Figure 3 for the CLPA path number 10 in Molines-en-Queyras township.

The data are updated every year through an annual CLPA campaign which usually corresponds to a calendar year. A typical data acquisition workflow is shown on an activity diagram in Figure
4. During a campaign, the CLPA inquiry is organized to collect data which allow reshaping the maximum limits of avalanche paths. The data are based on direct observation of avalanches, terrain analysis and communication received from external observers as ski resort professionals, rescue services, mountain guides, local habitants, etc. If required, these data can be completed by aerial photo analysis realized by a photo-interpreter. Eyewitness account and aerial photo-interpretation
techniques are used independently and plotted on the CLPA map as layers of different colours.

An observer collects available data for each CLPA avalanche path in zone under his responsibility. The data for a given path are summarized into a so-called CLPA sheet (Figure 5). On the basis of the collected evidences, the observer decides whether the existing avalanche limits should be extended on the CLPA map or not. Thus, a CLPA sheet supplies a complementary textual description of a path
to its geographical limits. It contains the following details:

– information about the inquiry (type, opening and closure dates, investigation zone);

– information about the studied avalanche path (path number, path label, location, ski resorts, corresponding EPA paths, protection structures);

– proof (or testimony) of an avalanche study, which contains an avalanche description in text
format and should be based on at least one source: a witness, a photograph or an archival document;

– identification number (idGis) of the geographic object corresponding to the avalanche path limits in the geographic information system.

The CLPA manager controls proposed modifications of avalanche limits, validates the new edition
of the CLPA and closes the inquiry. After the inquiry closure, no more modification can be applied to





CLPA sheets or avalanche limits. When the acquisition process is finished, the data distributor prepares an official distribution of the CLPA. He sends updated CLPA materials in paper format (maps, sheets, mountain massif description, etc.) to all stakeholders concerned by the current modification of avalanche limits. He also updates materials in numerical format available on the web site.

## 2.3 Classification of inhabited sites exposed to avalanches: SSA

The Sensitive Avalanche Paths (SSA) database is an inventory of 1 431 inhabited sites exposed to snow avalanches in France. The SSA inventory classifies the dangerousness of each site accordingly to three categories: strong, doubtful and low. The SSA concerns only the avalanche paths threatening inhabited buildings during winter seasons. The data acquisition for the inventory has been performed from 2003 to 2009 by the French National Forest Office following the method developed at Irstea by Rapin et al. (2006). No new data are collected at present. The data contains detailed information about the vulnerability, morphology, history and snow-climatology of each SSA site and provide a complementary information to the EPA and CLPA datasets. The SSA data are useful during high risk situations, for example, to prioritize evacuation of people living at most exposed locations.

## 3 User needs and system requirements

In order to build a web-based information system for snow avalanche observations, we first identified the user needs via a requirement analysis based on different methods as interviews (e.g. Burnay et al., 2014), scenario examples and observation of the working process in real conditions. These methods helped us to define system specifications that meet the expectations of the users. We then produced conceptual data models for EPA, CLPA and SSA systems which describe major entity in the avalanche observation workflow, and the relationships between these entities (Bourova et al., 2014). Finally, we used the conceptual data model and system specifications to create a database model and to design the architecture of the generalized system.

We defined the specifications of the avalanche information system through regular meetings with the EPA, CLPA and SSA managers and observers. One of the most important requirements was the workflow homogenisation for the observers from different affiliations (Irstea, French National Forest Office, partners). In the past, the observers from the French National Forest Office used to send a paper report to a manager in Irstea who copied the obtained data into an Access database via a desktop application. On the other hand, the observers from Irstea performed the data entry in a local copy of the Access database that was later synchronized with the central database. The data were then duplicated many times. Any modification in previous data by an observer resulted in data inconsistency among the local copies and central database. To avoid these problems, the new web-based system was designed to provide a homogeneous workflow for all the observers. It should manage multiple access by different actors and guarantees the data consistency at any time.


Another major requirement concerned the interoperability among different datasets. Historically separated, the EPA, CLPA and SSA datasets contain complementary data which are interesting to be explored together. The lack of interoperability was a real handicap for observers who tried to simultaneously exploit different data, continuously updated in real time.

    We had also to take into account that the observation of snow avalanches in France is a long

term project. The observation techniques became more and more efficient and this means that the working process, observed parameters and technical solutions constantly evolved over time and still continue to evolve (Bonnefoy-Demongeot et al., 2014). The system should be able to acknowledge past changes and also to easily accept possible modifications in several years even though theses future changes are completely unknown at present.

In terms of technical specifications, these different needs require that the new system has a robust, modular and highly decoupled architecture allowing easy modification and reuse of its components. We noted that the existing EPA-SSA application did not meet, unfortunately, these requirements. Therefore, we could not rely on the existing architecture but had better to build a new common framework for the CLPA and then adapt the EPA-SSA application accordingly.

## 215  4   System architecture

### 4.1   Main components

The new snow avalanche web system is composed of the three main components shown in Figure 6:

- a front web portal *http://www.avalanches.fr*;

- a webmapping tool (Deschatres et al., 2010a) for visualisation of EPA, CLPA and SSA geospa-
tial data *http://map.avalanches.fr*;

- a web application for entry and management of avalanche observations *http://extranet.avalanches.fr*.

    The front portal and the webmapping tool provide public access to all output materials issued from the EPA, CLPA and SSA observation workflow: avalanche maps, eyewitness reports, list of avalanche events, dangerousness estimation for inhabited sites exposed to avalanches. The web ap-
plication *http://extranet.avalanches.fr* was developed to support the observers in their daily work of data acquisition. It provides data entry, management and visualization in order to produce the output public data. The access to this web application is restricted to avalanche observers and protected by user login and password.

    The observed EPA, CLPA and SSA data are stored in a PostgreSQL relational database. The use of
a common database enables the interoperability of three datasets. The database stores also all technical information describing the business workflow: actors of the system, their corresponding roles and possible actions, data access rules and so on. The digitized geographic data for all datasets are





stored in format of polygons in a spatial database, implemented with ESRI ArcGIS geodatabase. The identification number idGis of each geographic object is stored in the main database to interconnect observations and corresponding output maps.

## 4.2 Application layers

The main objective of the web application *http://extranet.avalanches.fr* is to provide a flexible and reusable platform to manage observed data. In order to accept any possible modification in the future and to simplify the maintainability of the platform, we implemented a loosely coupled architecture mainly based on free and open source products as well as on publicly accessible international standards. The system is designed as a series of independent modules based on a stable framework. This approach allows us to reuse as much as possible the existing blocks while adding or modifying any module. The web application has been designed using Apache Tomcat servlet container and Java Enterprise Edition computing platform.The overall code is organized according to the model-view-controller design pattern (Buschmann et al., 1996). It is written in Java programming language using Spring, Spring MVC and Hibernate frameworks.

The system has an *n*-tier architecture composed of three layers: a data layer, a business layer and a presentation layer (Figure 7). To enhance the reusability of various parts of the application, we implemented these levels as independent as possible according to the principle of the inversion of control (Martin, 1996) allowing to reduce the dependencies between objects. Inversion of control is natively provided by the Spring framework (Gupta and Govil, 2010). In this configuration, the data and business layers are composed of Java classes that implement corresponding interfaces.

The data layer contains the information about data, data access and data sources that are stored in a database. The Data Access Objects (DAO) encapsulate all access to the database. The DAO java classes implement the DAO java interfaces that define standard operations to be performed on model objects. The model objects store the data that are retrieved from the database by the DAO classes. The data layer is implemented within the Hibernate framework.

The business layer encodes the business logic, i.e. the rules of the data processing during the observation workflow. While receiving a request from the presentation layer, the business layer decides which kind of data should be retrieved from the data layer, performs an appropriate data treatment and then returns the obtained result back to the presentation layer. The business layer represents a series of service components in Spring framework.

The presentation layer aims at displaying dynamically generated information to the end users in a web browser. It is composed of several web interfaces. Each web interface corresponds to a specific action of a user (e.g. creation, modification, visualization, suppression and validation of entities and procedures). The presentation level is developed in SpringMVC framework using JSP, JavaScript, jQuery and Ajax technologies.



Different components of the system are glued together within the Spring framework. One of the main advantages of three-layer decomposition is that the working protocol can be defined once within the business layer and then shared by any component within the presentation layer. The processing rules can be easily modified in one place, and then be instantly available throughout the whole application without having to make corresponding changes in any of the other layers.

### 4.3 Data traffic

To collect information about snow avalanches, the observers generally need to perform a terrain study at remote sites, sometimes with a poor internet connection speed. This condition implies that the system should carefully manage the data traffic between a server and a client. In order to avoid excessive bandwidth use, the system performs a major part of the data treatment on the server side. On the client side, the user interfaces display only the minimum quantity of data required to realize an action. In the case where large data need to be retrieved from the server, we use asynchronous data transfer based on Ajax technology. The data can then be retrieved from the server in the background without disturbing the loaded page.

### 5 Study case: the CLPA

The web application *http://extranet.avalanches.fr* provides a series of graphical user interfaces to organize annual campaigns of data acquisition, to enter avalanche observations, to process and to validate observed data. The application dynamically generates the interfaces which correspond to user's profiles. A user can perform only the actions defined by his role in the information system. The total number of graphical user interfaces currently attain 70, for all roles and level accesses. Approximately half of them correspond to the CLPA working process. In this section, as an illustrated example of the whole system, we describe the implementation of the main actions of the CLPA workflow.

There are three access levels for interfaces: management, data entry and data visualization. The management level allows performing campaign management actions and staff administration. In the CLPA tool, this level is reserved to a CLPA manager. His role is to create a new investigation zone, to define inquiries herein, to create new actors and to assign them to inquiries. The CLPA manager is also in charge of the data validation when an inquiry is finished. The CLPA manager has full access to CLPA data: he can add, modify or delete any data at any time. An example of a management level interface for inquiry creation is shown in Figure 8.

The data entry level allows CLPA observers to enter and to modify their own observations, but prohibits modification of data provided by other observers. At this level, there is no access to modify any investigation zone or any inquiry. An observer can only add data to an existing inquiry previously created and assigned to him by the CLPA manager. Figure 9 shows an example where an observer is


creating a new CLPA sheet for the inquiry "Molines-en-Queyras" defined in Figure 8. In the CLPA path box, he selects an existing path or can create a new one. In our example, the existing path number 10 is chosen and its relevant information (path label, ski resorts, protection structures, EPA paths) is then displayed. The GIS box is used to connect the CLPA sheet to a geographic object of avalanche limits in ArcGis geodatabase via its identification number. Finally, the observer adds proofs and their sources. The text of eyewitness report and the number of sources are summarized on the interface.

The data visualization level is assigned to a data distributor, a photo-interpreter and other actors who need to consult collected data without any modification. At this access level, a data distributor can also download CLPA materials for an investigation zone that has been already updated and formatted for distribution to stakeholders. Furthermore, this level provides access to the public version of CLPA sheets and to the digital library of photographs and documents.

The digital library (Figure 10) has been developed to enhance the interoperability between CLPA and EPA observation systems. It contains graphics and documents used in both systems: photographs for CLPA proofs, photographs of CLPA paths, CLPA archival documents, photographs of EPA paths and other reference documents of EPA events such as post event photographs showing snow deposit, damages and so on. The library accepts several graphical and PDF formats and can be used by all EPA-CLPA-SSA members. For each file, the information about the title, file name, person who imported the file, date of file import and details of use (CLPA inquiry, EPA events and so on) is displayed. The same file in the library can be used simultaneously in CLPA or EPA systems.

## 6   Discussion and conclusion

A web-based snow avalanche information system has been successfully implemented to manage various avalanche observation data. The design of the system architecture based on many open source components and frameworks assures the flexibility and the productivity of the platform. The system is modular and then can be easily extended for future needs.

This paper has illustrated the main technical advantages of this innovative and comprehensive framework. A full interoperability has been achieved by integrating EPA, CLPA and SSA datasets into a common harmonized database where they are now directly interconnected. For example, photographs and documents in all system are now shared via the digital library.

The achieved interoperability between different data types offers to users new outlooks in their work. For instance, it technically allows connecting EPA events to an update of the CLPA map. It means that an EPA event can become a source for a CLPA proof while creating new CLPA sheets. Also, combined analysis with other increasingly detailed environmental (climate data, land use and land cover data, etc.) and socio-economic (position and types of buildings, critical infrastructures, road traffic densities, etc.) databases is now much easier, opening many promising new investiga-





tions fields. Specifically, temporal and spatial aspects of snow avalanche activity and risk can now be jointly explored much more efficiently taking into account their main drivers, which represents a crucial improvement both for scientific purposes and for local on site risk management by technicians and stakeholders. Already available examples having largely beneficiated of these new facilities include detailed investigation of climatic drivers of avalanche activity in the French Alps (Lavigne et al., 2015), long term forecasting of extreme avalanche events (Eckert et al., 2010b), cross comparison with tree ring data and connection to afforestation processes (Schlappy et al., 2014), short term forecasting of major winter storms (Dkengne et al., 2016), socio-economic assessment of the vulnerability of various types of buildings (Favier et al., 2014), and subsequent risk assessment for inhabited areas (Eckert et al., 2008).

The flexibility of the new architecture opens a perspective toward a full integrated open source platform. The only proprietary component of our system ArcGIS geodatabase is expected to migrate soon into an open source spatial database operated by PostgreSQL and PostGIS. The geographic information could then be accessed in a more interactive manner simplifying the working process for technical staff and for end users.

Our approach could certainly be transferred with benefits to avalanche records from other countries (with slight adaptations respecting their local specificities), and, more generally, to other natural hazards generating risk in France or elsewhere, and for which currently existing information systems no longer fulfill the requirements of various users.

## 7 Data access

The EPA, CLPA and SSA output materials are public. Even if the full management data access is restricted to the project members and partners via *http://extranet.avalanches.fr*, they can therefore be downloaded freely by anyone from the website *http://www.avalanches.fr*. In addition to the raw data, the website contains a huge amount of information about the data specificities, advices for potential users, additional data such as old maps and snowfall records, etc. Also, summaries of ongoing and past research projects grounding on the French snow avalanche information system are provided. For instance, an exhaustive bibliographic references list can be found at *http://www.avalanches.fr/liste-bibliographique*. All the material is regularly updated.

*Acknowledgements.* This work was supported by the French Ministry of the Environment. We are grateful to Thibault Degiuli, Ambroise Petitgenêt and Mickaël Bonilla for valuable discussions on the snow avalanche information system architecture. We would like also to thank Frederic Flin for reading a preliminary version of this manuscript and for making useful comments. Irstea is a part of Labex OSUG@2020 (ANR-10-LABX-56).





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




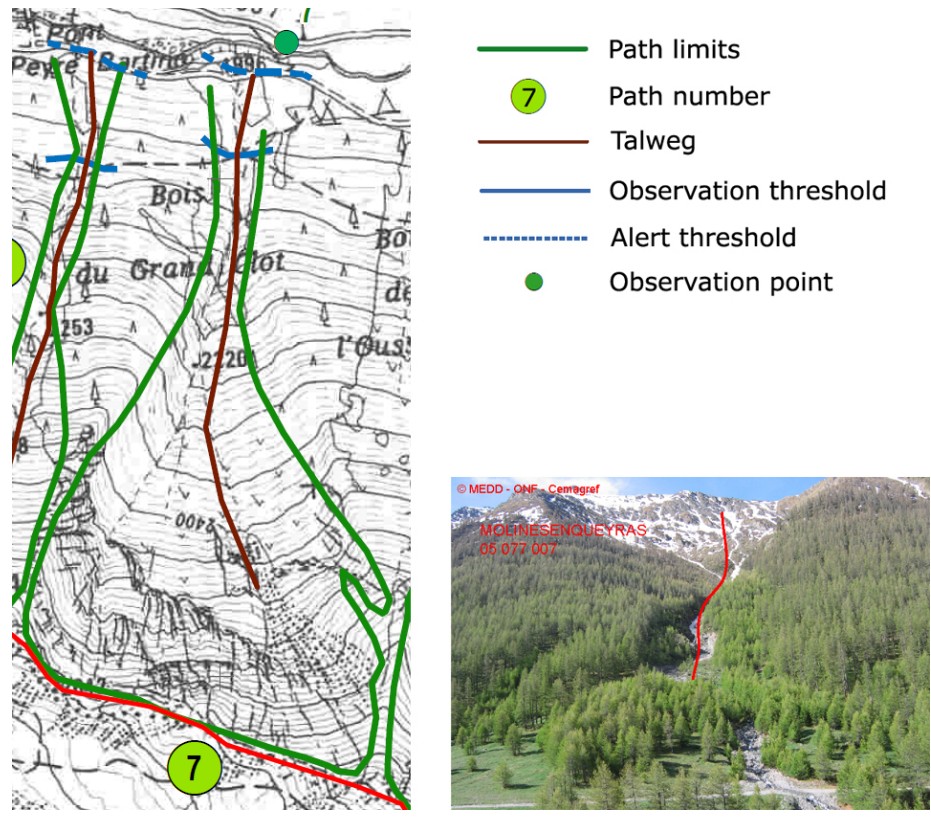

**Figure 1.** A map and a photograph of the EPA path number 7 in Molines-en-Queyras township. The observation threshold is plotted as a solid blue line. The alert threshold is drawn as a dashed blue line. For this path, the observation threshold is chosen near a large rock block while the alert threshold corresponds to a cross-country ski track. Both thresholds are well visible and can be easily identified by an observer from the observation point. The EPA database stores all avalanche events that reached at least the observation threshold.





**Figure 2.** The first page of a PDF document listing recent avalanche events observed on EPA path number 7 from Molines-en-Queyras township shown in Figure 1. In total, 24 events have been observed on this path since 1950.



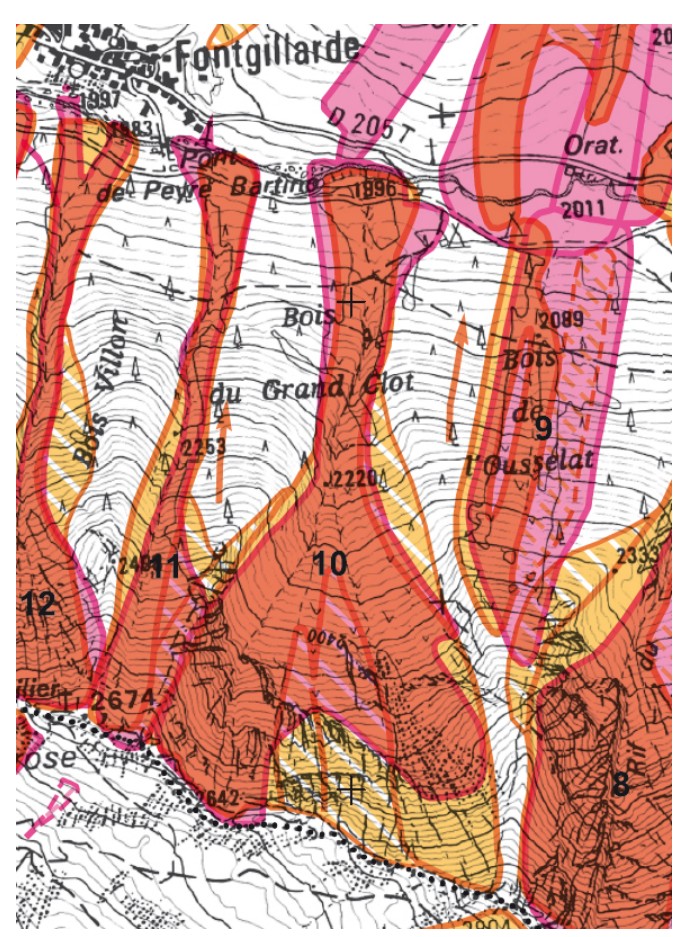

**Figure 3.** The Localization Map of Avalanche Phenomena (CLPA) in Molines-en-Queyras township. The CLPA path number 10 corresponds to the EPA path number 7 presented in Figure 1. The avalanche extension limits obtained from eyewitness accounts are plotted in magenta. The yellow contours show avalanche extension limits based on photo-interpretations. Orange areas represent the intersection of these two contours.





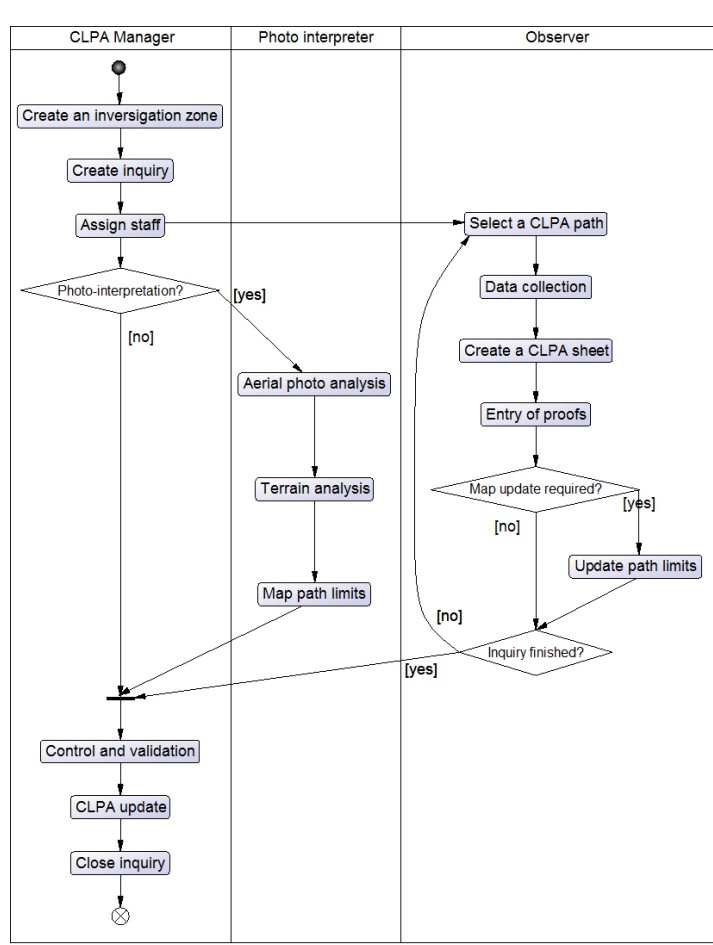

**Figure 4.** Activity diagram describing the process of data acquisition during a CLPA campaign.


| MEDDE-ONF-Irstea | CLPA Sheet | 05077 - 10 - 76361 |
|---|---|---|
| **Township** <br> MOLINES-EN-QUEYRAS 05077 | **Investigation zone** <br> Molines-En-Queyras 2010 | **Inquiry type** <br> Decennial update |
| **Inquiry** <br> Molines-En-Queyras | **Opening date** <br> 01/01/2010 | **Closure date** <br> 31/12/2010 |
| **Path number** <br> 10 | **Path label** <br> Combe Crose, Avalanche du Grand Bois | **Ski resort** |
| **Created on** <br> 01/09/2010 | **Map number** | |

**Eyewitness report**

- [doc. 1254] - [photos 4124, 4125] - [GAUCHER Romain]

Une avalanche s'est produite dans ce couloir, probablement lors de l'hiver 2008/2009. le phénomène a dépassé les limites de son emprise précédente. En rive droite, l'avalanche a dépassé de 80 mètres ses précédentes limites ainsi que sur sa partie frontale, une partie de l'écoulement est venue remonter sur le versant en face entre la route et le torrent.

- [photos 4124, 4125] - [BRUNET Daniel, DENIAU Dominique]

Les témoins évoquent une avalanche qui s'est produite après les chutes de neige importante de décembre 2008. Le dépôt était très important, traversait la rivière et venait se caler contre le talus de la route. Il pouvait atteindre une dizaine de mètre d'épaisseur. De nombreux arbres de toutes tailles ont été cassés. L'avalanche a dépassé les limites de l'emprise déjà connue.

**EPA paths**

007 BOIS DU GRAND CLOT

**Avalanche protection structures**

**Archive documents**

[ref 1254] Photographies Service RTM Gap, ONF, Avalanches de décembre 2008, Queyras (2008)

**Previous inquiries**

[Numerotation unchanged] Path n° 10, MOLINES-EN-QUEYRAS 05077, inquiry Queyras 01/01/2002

**Next inquiries**

[Numerotation unchanged] Path n° 10, MOLINES-EN-QUEYRAS 05077, inquiry Mise à jour annuelle 01/01/2011

**Notes**

Fiche signalétique. Enquête permanente ONF n°7.

**Pictures**

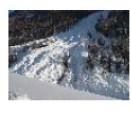 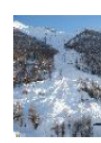

[ref 4124] 01/09/2010       [ref 4125] 01/09/2010

**Figure 5.** A CLPA sheet for the CLPA path number 10 in Molines-en-Queyras township. It contains eyewitness reports, archival documents, photographs and other information obtained from this path during a CLPA inquiry. Eyewitness reports are currently available in French only. Based on this information, an observer decides if the existing contour of the path needs to be updated or not. If necessary, the observer draws a new contour (see Figure 3).



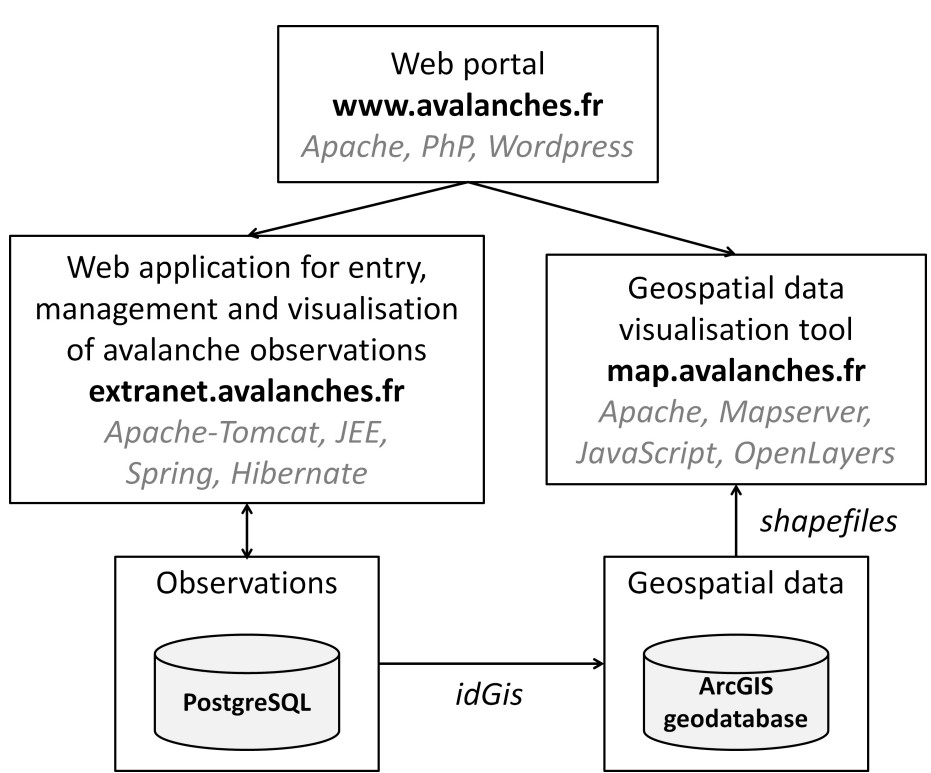

**Figure 6.** Main components of the information system.




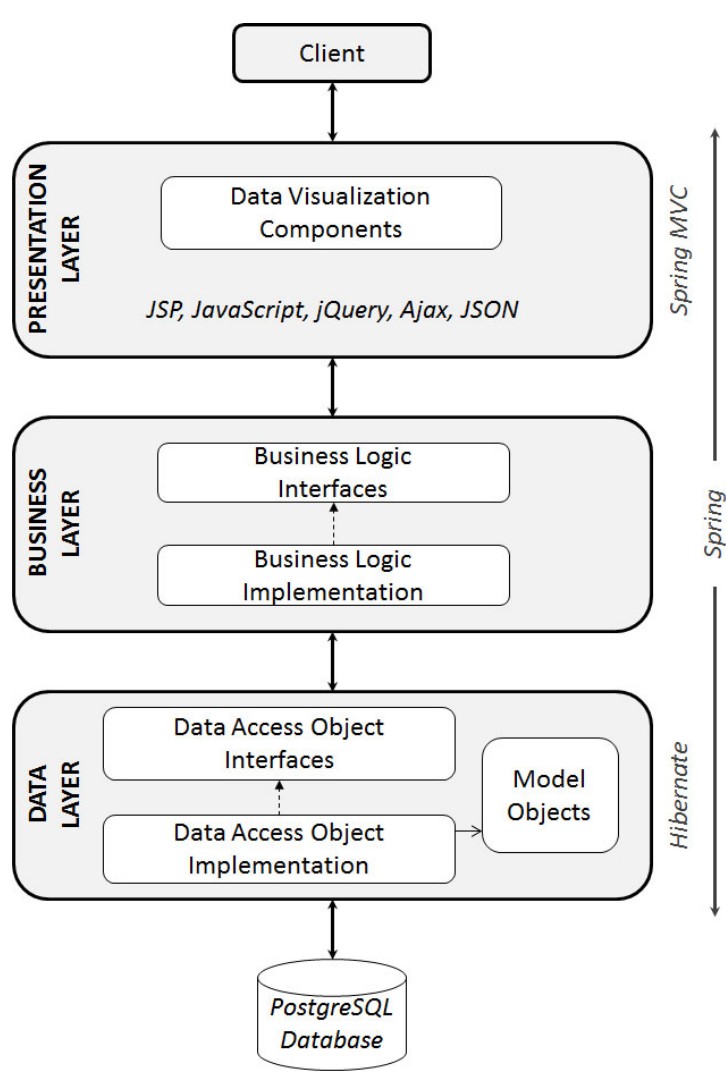

**Figure 7.** Architecture of the web application for entry and management of avalanche observations.





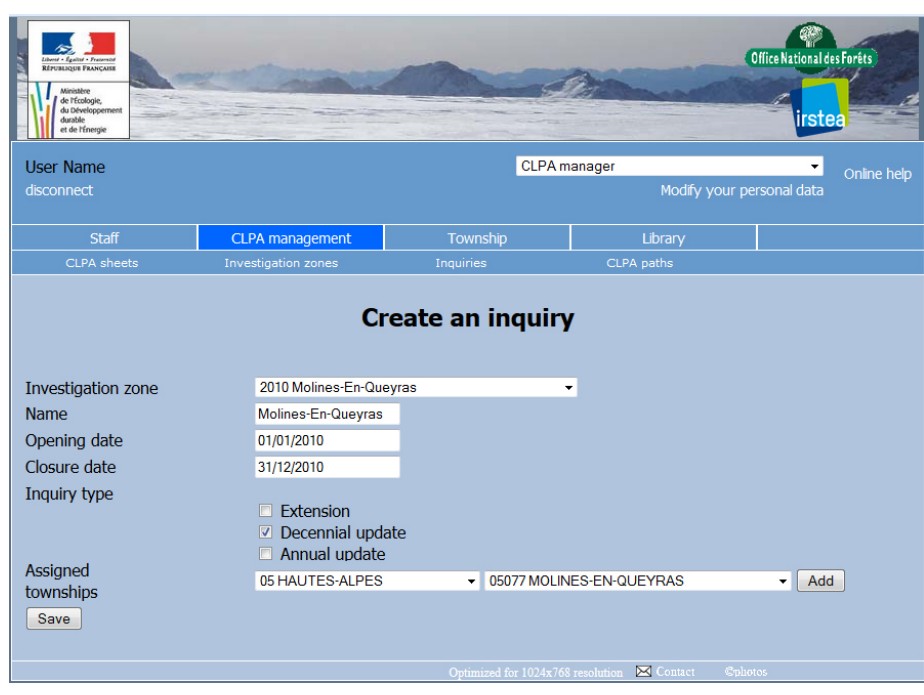

**Figure 8.** Graphical user interface of the management level to create a CLPA inquiry. An inquiry named "Molines-en-Queyras" is being created on an existing investigation zone. The inquiry time period, inquiry type and assigned township are defined.







**Figure 9.** Graphical user interface of the data entry level to create a CLPA sheet.





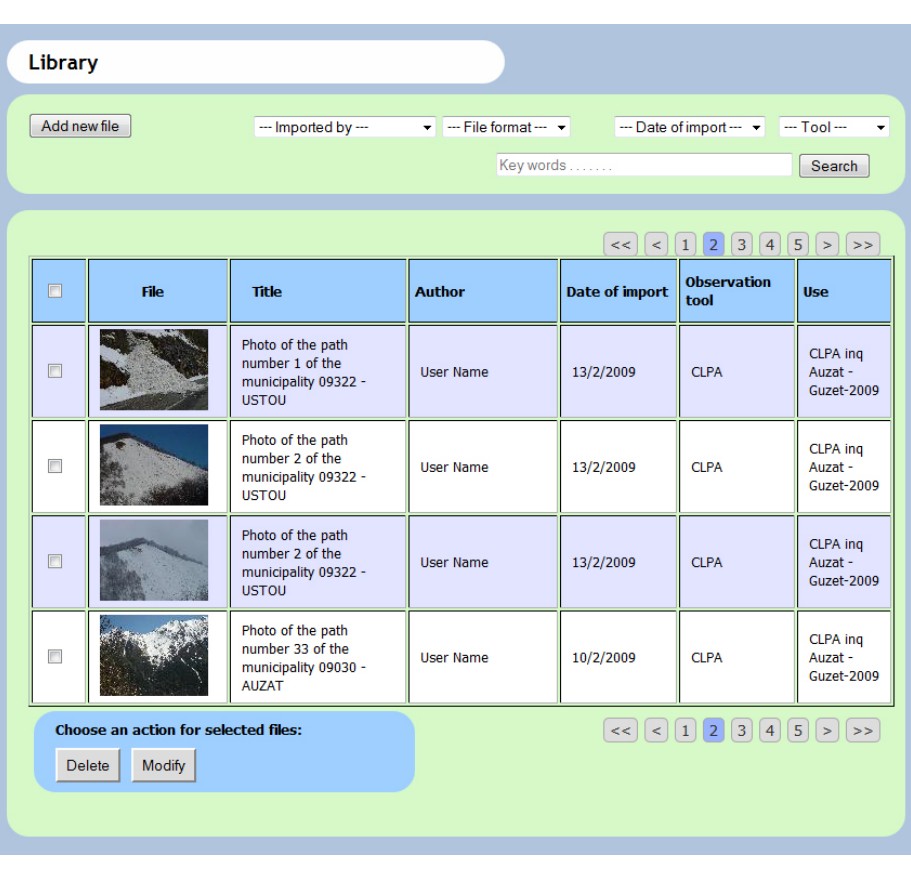

**Figure 10.** Digital library shared by the CLPA and EPA datasets.