# Peer review of "A new web-based system to improve the monitoring of snow avalanche hazard in France"

_Natural Hazards and Earth System Sciences, 2015_

## Referee Comment (RC1) · B. Zweifel (Referee) · 20 Feb 2016

General comment

Dear authors, first of all I would like to congratulate you for this very concise article on a web-based system to improve avalanche monitoring in France. In my eyes, the article is very well written and clearly understandable for a large community in avalanche safety research. However, I have significant concerns if this article really belongs to a research journal since its topic is clearly in applied research. Nevertheless, I would encourage the editor of the journal to publish this highly relevant article for the avalanche community. Several other research institutions might face some similar problematics and a share of knowledge in this field might be very valuabel.

Specific comments and technical corrections

[Figure]

As I mentioned above, I find the article very well written and consequently I have only some minor comments: - Line 30: please use the correct term for the SLF institute which is: WSL Institute for Snow and Avalanche Research SLF - Line 32 to 34: I would recommend to use a more up-to-date publication of the DADB (I would suggest this publication: Techel, F., & Zweifel, B. (2013). Recreational avalanche accidents in Switzerland: Trends and patterns with an emphasis on burial, rescue methods and avalanche danger. In F. Naaim-Bouvet, Y. Durand & R. Lambert (Eds.), International Snow Science Workshop 2013, Proceedings (pp. 1106-1112). Grenoble, France: ANENA, IRSTEA, Météo-France.) By the way: DADB is there mentioned as ADB. However, I think, both is fine since there is no official English term for this database. - Line 76: Is there any specific reason to write Access with a capital letter? - Line 104, 129: I think, peer-reviewed is not the appropriate term for the procedure you mention. It is more of a cross check through another person. The process of peer-review is closely correlated to scientific writing though. - Line 125: I'm just wondering: 48 h is quite a lot since conditions can change very quickly especially in these storm periods when the significant avalanche happen. Is there any reason for taking that long time or would there be the chance to be quicker? - Line 136: the term i.e. seems to be in a different format? - Section 4: I have to honest that this is not at all my knowledge area. So, I would recommend the editor to have at least one reviewer for this article with a software developer background.
* * *

---

## Referee Comment (RC2) · S. Harvey (Referee) · 15 Mar 2016

Summary:

The paper reports the development of a web-based snow avalanche information system. It merges different avalanche datasets like avalanche chronicle (EPA), avalanche maps (CLPA) and a classification of inhabited sites exposed to avalanches (SSA) into one common database. The system enables to operate between these different types of data like maps, tables, photos and descriptive parameters. With modular components it can be used for data entry, data search and offers a public open access to avalanche data.

Key elements of judgment:

The presented web-based system is an innovative approach to bring all sorts of avalanche data together and also to use it for data entry and as avalanche management tool among avalanche paths. What is different to existing web-based systems in other countries is, that is focuses on avalanche paths and combines observations with avalanche hazard maps as well as avalanche occurrence. A great advantage is that photos and documents of avalanche events can be stored and are available in the digital library. Further choosing an open source solutions as much as possible makes the platform adaptable. I also think it is a great benefit that a lot of data and information is available to the public. Nevertheless the paper does not really present novel concepts in management of avalanche data. In my opinion the paper is not really scientific. It is more a description on how different avalanche data are implemented into a web-based system to monitor and evaluate avalanche paths.

Data collection and modification of the avalanche map needs clear guidelines. The authors do not really explain how collected data is peer-reviewed. Further they mention a requirement analysis but the paper does not explain to what kind of users this new platform mainly focuses.

Working through the webpage is not so intuitive and need some introduction. Looking at specific avalanche path on the map, I could not find any further description of past avalanches or specific object data. It seams obvious to click on an avalanche path to get more information, but nothing happens there, at least on the public access.

Overall, I think the authors' work is a worthwhile contribution and should be published.

---

## Author Comment (AC1) · 18 Apr 2016

Dear Dr. Zweifel,

We are very grateful for your encouragements and your valuable comments. We completely agree that this article essentially concerns the applied research field. Our objective was to develop a new automated system to assist observers in their daily monitoring activity, enhancing the quality and consistency of obtained data. The data from different sources are now stored in a common database and could then be easily explored by researchers. Thus, the new platform facilitates innovative developments to better understand avalanche activity. We believe that this approach could be transferred to other countries and/or other related natural hazards and risks.

Please, find detailed comment-by-comment answers in what follows:

[Figure]

Comment: Line 30: please use the correct term for the SLF institute which is: WSL Institute for Snow and Avalanche Research SLF

Reply: We modified the SLF institute name as suggested.

Comment: Line 32 to 34: I would recommend to use a more up-to-date publication of the DADB (I would suggest this publication: Techel, F., & Zweifel, B. (2013). Recreational avalanche accidents in Switzerland: Trends and patterns with an emphasis on burial, rescue methods and avalanche danger. In F. Naaim-Bouvet, Y. Durand & R. Lambert (Eds.), International Snow Science Workshop 2013, Proceedings (pp. 1106-1112). Grenoble, France: ANENA, IRSTEA, Météo-France.) By the way: DADB is there mentioned as ADB. However, I think, both is fine since there is no official English term for this database.

Reply: We included the cited reference in the text and specified that the DADB is also named ADB.

Comment: Line 76: Is there any specific reason to write Access with a capital letter?

Reply: Here, "Access" is the name of the software used in a previous version of our platform. To avoid any misunderstanding with the common word "access", we changed "Access" to "Microsoft Access" in the text.

Comment: - Line 104, 129: I think, peer-reviewed is not the appropriate term for the procedure you mention. It is more of a cross check through another person. The process of peer-review is closely correlated to scientific writing though.

Reply: The "cross check" term is indeed much more accurate in this situation. Thank you for this suggestion. We modified the text accordingly.

Comment: Line 125: I'm just wondering: 48 h is quite a lot since conditions can change very quickly especially in these storm periods when the significant avalanche happen. Is there any reason for taking that long time or would there be the chance to be quicker?

[Figure]

Reply: For major avalanches that attain the alert threshold, the observer is expected to alert Irstea and the National Forest Office for possible further research investigations. For example, to collect data about the snow deposit of such exceptional events could be particularly important for CLPA map updates. The observer is asked to rise alert as soon as possible. 48 hours are given in the guidelines as a reasonable maximum limit.

Comment: - Line 136: the term i.e. seems to be in a different format?

Reply: We corrected this issue in the text.

---

## Author Comment (AC2) · 18 Apr 2016

Dear Dr. Harvey,

First of all, we would like to thank you for helpful comments on the manuscript and feed-backs on the avalanche web platform. We adapted the text of the paper accordingly to these comments. The feedbacks on the web application are precious for us. They suggest several improvements that we expect to address rapidly in future versions of the platform. Most of these issues have been already proposed to the French Ministry of the Environment in the frame of next milestones in this project.

Point-by-point answers:

Comment: Nevertheless, the paper does not really present novel concepts in manage-

Interactive
comment

ment of avalanche data. In my opinion the paper is not really scientific. It is more a description on how different avalanche data are implemented into a web-based system to monitor and evaluate avalanche paths.

Reply: We agree that the novelty of the paper doesn't concern research results in snow avalanches. The objective of our work was to develop a new modern tool for avalanche data management that contributes to research advancements. The new platform improves the consistency and quality of collected data by providing an assisted workflow for avalanche observers. It helps researchers to access and to jointly exploit data of different sources. We believe that our approach can be transferred to avalanche records from other countries and can be helpful to other researchers and engineers in snow sciences.

Comment: Data collection and modification of the avalanche map needs clear guidelines.

Reply: In section 2.2 the paper describes a typical procedure of data acquisition for the avalanche map (CLPA) protocol. The details of the CLPA protocol has already been published previously. The corresponding references have been given in Introduction. Following your comment, we adapted the text of section 2.2 by explaining clearly this point with relevant bibliographical references.

Comment: The authors do not really explain how collected data is peer-reviewed.

Reply: Dr. Zweifel, a referee for this manuscript, pointed that the term "peer-reviewed" would not be appropriate for described procedure because the process of peer-review is mostly correlated to scientific writing. The "cross-check" term is more accurate. We thus changed the term in the text. In section 2.2 describing the CLPA protocol, we added an explanation of the cross-check procedure. In section 2.1 describing the EPA protocol, we added a relevant bibliographical reference that describes the cross-check procedure.

Comment: Further they mention a requirement analysis but the paper does not explain to what kind of users this new platform mainly focuses.

Reply: In section 3, we added a description of the end-users mainly focused by the new platform.

Comment: Working through the webpage is not so intuitive and need some introduction. Looking at specific avalanche path on the map, I could not find any further description of past avalanches or specific object data. It seems obvious to click on an avalanche path to get more information, but nothing happens there, at least on the public access.

Reply: The construction of the avalanche platform follows consecutive steps, annually planned and approved by the French Ministry of the Environment. The platform is aimed at several end-user groups with different needs. We searched a compromise to address the needs of these users, depending on available resources. The needs of observers, researchers and stakeholders have been prioritized in order to enable consistency, quality and interoperability of collected data and to provide full data access for concerned users. This access is reserved to project members and partners.

We are now working on improving data access for general public, in French and also in English. Recently, we renewed output materials to benefit from EPA-CLPA-SSA data interoperability provided by the new platform. The materials are publically available in French version. An English version of the portal is under construction. For the moment, we translated main output materials in English. They are available from direct links but not yet related to the main portal:

Lists of EPA events:

ftp://avalanchesftp.grenoble.cemagref.fr/epaclpa/EPA_listes_evenements_EN/

List of CLPA sheets:

ftp://avalanchesftp.grenoble.cemagref.fr/epaclpa/CLPA_fiches_signaletiques_EN/

Finally, we are also planning to connect geographic maps to textual description of past avalanches by direct links, as you suggest. To implement this, we need first to migrate geographic data from the actual ArcGIS database into a common database containing textual and geographic information under PostgreSQL/PostGIS. These developments are scheduled in future versions of the platform.

The above points are summarized in Section 6 of the paper.
* * *